# Application of Artificial Intelligence in Pathology: Trends and Challenges

**DOI:** 10.3390/diagnostics12112794

**Published:** 2022-11-15

**Authors:** Inho Kim, Kyungmin Kang, Youngjae Song, Tae-Jung Kim

**Affiliations:** 1College of Medicine, The Catholic University of Korea, 222 Banpo-daero, Seocho-gu, Seoul 06591, Republic of Korea; 2Department of Hospital Pathology, Yeouido St. Mary’s Hospital, College of Medicine, The Catholic University of Korea, 10, 63-ro, Yeongdeungpo-gu, Seoul 07345, Republic of Korea

**Keywords:** artificial intelligence, computational pathology, digital pathology, histopathology image analysis, deep learning

## Abstract

Given the recent success of artificial intelligence (AI) in computer vision applications, many pathologists anticipate that AI will be able to assist them in a variety of digital pathology tasks. Simultaneously, tremendous advancements in deep learning have enabled a synergy with artificial intelligence (AI), allowing for image-based diagnosis on the background of digital pathology. There are efforts for developing AI-based tools to save pathologists time and eliminate errors. Here, we describe the elements in the development of computational pathology (CPATH), its applicability to AI development, and the challenges it faces, such as algorithm validation and interpretability, computing systems, reimbursement, ethics, and regulations. Furthermore, we present an overview of novel AI-based approaches that could be integrated into pathology laboratory workflows.

## 1. Introduction

Pathologists examine pathology slides under a microscope. To diagnose diseases with these glass slides, many traditional technologies, such as hematoxylin and eosin (H&E) staining and special staining, have been used. However, even for experienced pathologists, intra- and interobserver disagreement cannot be avoided through visual observation and subjective interpretation [1]. This limited agreement has resulted in the necessity of computational methods for pathological diagnosis [2,3,4]. Because automated approaches can achieve reliable results, digital imaging is the first step in computer-aided analysis [5]. When compared to traditional digital imaging technologies that process static images through cameras, whole-slide imaging (WSI) is a more advanced and widely used technology in pathology [6].

Digital pathology refers to the environment that includes tools and systems for digitizing pathology slides and associated metadata, in addition their storage, evaluation, and analysis, as well as supporting infrastructure. WSI has been proven in multiple studies to have an excellent correlation with traditional light microscopy diagnosis [7] and to be a reliable tool for routine surgical pathology diagnosis [8,9]. Indeed, WSI technology provides a number of advantages over traditional microscopy, including portability, ease of sharing and retrieving images, and task balance [10]. The establishment of the digital pathology environment contributed to the development of a new branch of pathology known as computational pathology (CPATH) [11]. Novel terminology and definitions have resulted from advances in computational pathology (Table 1) [12]. The computational analysis of pathology slide images has made direct disease investigation possible rather than relying on a pathologist analyzing images on a screen [13]. AI approaches aided by deep learning results are frequently used to combine information from digitized pathology images with their associated metadata. Using AI approaches that computationally evaluate the entire slide image, researchers can detect features that are difficult to detect by eye alone, which is now the state-of-the-art in digital pathology [14].

The conventional pathological digital image machine learning method requires particularly educated pathologists to manually categorize abnormal picture attributes before incorporating them into algorithms. Manually extracting and analyzing features from pathological images was a time-consuming, labor-intensive, and costly method that led to many disagreements among pathologists on whether features are typical [15]. Human-extracted visual characteristics must be translated into numerical forms for computer algorithms, but identifying patterns and expressing them with a finite number of feature markers was nearly impossible in some complex diseases. Diverse and popular studies to ‘well’ learn handmade features became the basis for a commercially available medical image analysis system. After all the algorithm development steps, its performance often had a high false-positive rate, and generalization in even typical pathological images was limited [16]. Deep learning, however, has enabled computers to automatically extract feature vectors from pathology image example data and learn to build optimal algorithms on their own [17,18], even outperforming physicians in some cases, and has now emerged as a cutting-edge machine learning method in medical clinical practice [19]. Diverse deep architectures trained with huge image datasets provide biological informatics discoveries and outstanding object recognition [20].

The purpose of this review is to enhance the understanding of the reader with an update on the implementation of artificial intelligence in the pathology department regarding requirements, work process and clinical application development.

## 2. Deveopment of AI Aided Computational Pathology

Integrating artificial intelligence into the workflow of the pathology department can perform quality control of the pre-analytic, analytic, and post-analytic phases of the pathology department’s work process, allowing quality control of scan images and formalin-fixed paraffin-embedded tissue blocks, integrated diagnosis with joining clinical information, ordering necessary pathology studies including immunohistochemistry and molecular studies, automating repetitive tasks, on-demand consultation, and cloud server management (Figure 1), which, finally allow precision medicine by enabling us to use a wide range of patient data, including pathological images, to develop disease-preventive and treatment methods tailored to individual patient features. To achieve the above-mentioned goals, there are crucial elements required for CPATH. A simple summary of the required steps for the application of an AI with CPATH is demonstrated in Figure 2.

### 2.1. Equipment

Transitioning from glass to digital workflows in AP requires new digital pathology equipment, image management systems, improved data management and storage capacities, and additional trained technicians [21]. While the use of advanced high-resolution hardware with multiple graphical processing units can speed up training, it can become prohibitively expensive. Pathologists must agree to changes to a century-old workflow. Given that change takes time, pathologist end-users should anticipate change-management challenges independent of technological and financial hurdles. AI deployment in the pathology department requires digital pathology. Digital pathology has many proven uses, including primary and secondary clinical diagnosis, telepathology, slide sharing, research data set development, and pathology education or teaching [22]. Digital pathology systems provide time- and cost-saving improvements over the traditional microscopy technique and improve inter-observer variation with adequate slide image management software, integrated reporting systems, improved scanning speeds, and high-quality images. Significant barriers include the introduction of technologies without regulatory-driven, evidence-based validation, the resistance of developers (academic and industrial), and the requirement for commercial integration and open-source data formats.

### 2.2. Whole Slide Image

In the field of radiology, picture archiving and communication systems (PACS) were successfully introduced owing to infrastructure such as stable servers and high-performance processing devices, and they are now widely used in deep learning sources [23,24]. Similarly, in the pathology field, a digital pathology system was developed that scans traditional glass slides using a slide scanner to produce a WSI; it then stores and transmits it to servers [13]. Because WSI which has an average of 1.6 billion pixels and occupies 4600 megabytes (MB) per unit, thus taking up much more space than a DICOM (digital imaging and communications in medicine) format, this technique took place later in pathology than in radiography [25]. However, in recent years, scanners, servers, and technology that can quickly process WSI have made this possible, allowing pathologists to inspect images on a PC screen [6].

### 2.3. Quality Control Using Artificial Intelligence

AI tools can be embedded within a pathology laboratory workflow before or after the diagnosis of the pathologist. Before cases are sent to pathologists for review, an AI tool can be used to triage them (for example, cancer priority or improper tissue section) or to help with screening for unexpected events (e.g., tissue contamination or microorganisms). After reviewing a case, pathologists can also use AI tools to execute certain tasks (e.g., counting mitotic figures for tumor grading or measuring nucleic acid quantification). AI software can also run in the background and execute tasks such as quality control and other tasks all the time (e.g., correlation with clinical or surgical information). The ability of AI, digital pathology, and laboratory information systems to work together is the key to making a successful AI workflow that fits the needs of a pathology department. Furthermore, pre-analytic AI implementation can affect the process of molecular pathology. Personalized medicine and accurate quantification of tumor and biomarker expression have emerged as critical components of cancer diagnostics. Quality control (QC) of clinical tissue samples is required to confirm the adequacy of tumor tissue to proceed with further molecular analysis [26]. The digitization of stained tissue slides provides a valuable way to archive, preserve, and retrieve important information when needed.

### 2.4. Diagnosis and Quantitation

A combination of deep learning methods in CPATH has been developed to excavate unique and remarkable biomarkers for clinical applications. Tumor-infiltrating lymphocytes (TILs) are a prime illustration, as their spatial distributions have been demonstrated to be useful for cancer diagnosis and prognosis in the field of oncology [27]. TILs are the principal activator of anticancer immunity in theory, and if TILs could be objectively measured across the tumor microenvironment (TME), they could be a reliable biomarker [20]. TILs have been shown to be associated with recurrence and genetic mutations in non-small cell lung cancer (NSCLC) [28], and lymphocytes, which have been actively made immune, have proved to have a better response, leading to a longer progression-free survival than the ones that did not show much immunity [29]. Because manual quantification necessitates a tremendous amount of work and is easily influenced by interobserver heterogeneity [30,31], many approaches are being tested in order to overcome these hurdles and determine a clinically meaningful TIL cutoff threshold [32]. Recently, a spatial molecular imaging technique obtaining spatial lymphocytic patterns linked to the rich genomic characterization of TCGA samples has exemplified one application of the TCGA image archives, providing insights into the tumor-immune microenvironment [20].

On a cellular level, spatial organization analysis of TME containing multiple cell types, rather than only TILs, has been explored, and it is expected to yield information on tumor progression, metastasis, and treatment outcomes [33]. Tissue segmentation is done using the comprehensive immunolabeling of specific cell types or spatial transcriptomics to identify a link between tissue content and clinical features, such as survival and recurrence [34,35]. In a similar approach, assessing image analysis on tissue components, particularly focusing on the relative amount of area of tumor and intratumoral stroma, such as the tumor-stroma ratio (TSR), is a widely studied prognostic factor in several cancers, including breast cancer [36,37], colorectal cancer [38,39], and lung cancer [40]. Other studies in CPATH include an attempt to predict the origin of a tumor in cancers of unknown primary source using only a histopathology image of the metastatic site [41].

One of the advantages of CPATH is that it allows the simultaneous inspection of histopathology images along with patient metadata, such as demographic, gene sequencing or expression data, and progression and treatment outcomes. Several attempts are being made to integrate patient pathological tissue images and one or more metadata to obtain novel information that may be used for diagnosis and prediction, as it was discovered that predicting survival using merely pathologic tissue images was challenging and inaccurate [42]. Mobadersany et al. used a Cox proportional hazards model integrated with a CNN to predict the overall survival of patients with gliomas using tissue biopsy images and genetic biomarkers such as chromosome deletion and gene mutation [43]. He et al. used H&E histopathology images and spatial transcriptomics, which analyzes RNA to assess gene activity and allocate cell types to their locations in histology sections to construct a deep learning algorithm to predict genomic expression in patients with breast cancer [44]. Furthermore, Wang et al. employed a technique known as ‘transcriptome-wide expression-morphology’ analysis, which allows for the prediction of mRNA expression and proliferation markers using conventional histopathology WSIs from patients with breast cancer [45]. It is also highly promising in that, as deep learning algorithms progress in CPATH, it can be a helpful tool for pathologists and doctors making decisions. Studies have been undertaken to see how significant an impact assisting diagnosis can have. Wang et al. showed that pathologists employing a predictive deep learning model to diagnose the metastasis of breast cancer from WSIs of sentinel lymph nodes reduced the human error rate by nearly 85% [46]. In a similar approach, Steiner et al. looked at the influence of AI in the histological evaluation of breast cancer with lymph node metastasis, comparing pathologist performance supported by AI with pathologist performance unassisted by AI to see whether supplementation may help. It was discovered that algorithm-assisted pathologists outperformed unassisted pathologists in terms of accuracy, sensitivity, and time effectiveness [47].

## 3. Deep Learning from Computational Pathology

### 3.1. International Competitions

The exponential development in scanner performance making producing WSI easier and faster than previously, along with sophisticated viewing devices, major advancements in both computer technology and AI, as well as the accordance to regulatory requirements of the complete infrastructure within the clinical context, have fueled CPATH’s rapid growth in recent years [15]. Following the initial application of CNNs in histopathology at ICPR 2012 [48], several studies have been conducted to assess the performance of automated deep learning algorithms analyzing histopathology images in a variety of diseases, primarily cancer. CPATH challenges are being promoted in the same way that competitions and challenges are held in the field of computer engineering to develop technologies and discover talented rookies. CAMELYON16 was the first grand challenge ever held, with the goal of developing CPATH solutions for the detection of breast cancer metastases in H&E-stained slides of sentinel lymph nodes and to assess the accuracy of the deep learning algorithms developed by competition participants, medical students and experienced professional pathologists [49]. The dataset from the CAMELYON16 challenge, which took a great deal of work, was used in several other studies and provided motive for other challenges [50,51,52], attracting major machine learning companies such as Google to the medical artificial intelligence field [53], and is said to have influenced US government policy [54]. Since then, new challenges have been proposed in many more cancer areas using other deep learning architectures with greater datasets, providing the driving force behind the growth of CPATH (Table 2). Histopathology deep learning challenges can attract non-medical engineers and medical personnel, provide prospects for businesses, and make the competition’s dataset publicly available, benefiting future studies. Stronger deep learning algorithms are expected to emerge, speeding the clinical use of new algorithms in digital image analysis. Traditional digital image analysis works on three major types of measures: image object localization, classification, and quantification [12], and deep learning in CPATH focuses on those metrics similarly. CPATH applications include tumor detection and classification, invasive or metastatic foci detection, primarily lymph nodes, image segmentation and analysis of spatial information, including ratio and density, cell and nuclei classification, mitosis counting, gene mutation prediction, and histological scoring. Two or more of these categories are often researched together, and deep learning architectures like convolutional neural networks (CNN) and recurrent neural networks are utilized for training and applications.

### 3.2. Dataset and Deep Learning Model

Since public datasets for machine learning learning in CPATH, such as the Cancer Genome Atlas (TCGA), the Cancer Image Archive (TCIA), and public datasets created by several challenges, such as the CAMELYON16 challenge dataset, are freely accessible to anyone, researchers who do not have their own private data can conduct research and can also use the same dataset as a standard benchmark by several researchers comparing the performance of each algorithm [15]. Coudray et al. [66], Using the inception-v3 model as a deep learning architecture, assessed the performance of algorithms in classification and genomics mutation prediction of NSCLC histopathology pictures from TCGA and a portion of an independent private dataset, which was a noteworthy study that could detect genetic mutations using WSIs such as STK11 (AUC 0.85), KRAS (AUC 0.81), and EGFR (AUC 0.75). Guo et al. used the Inception-v3 model to classify the tumor region of a breast cancer [67]. Bulten et al. used 1243 WSIs of private prostate biopsies, segmenting individual glands to determine Gleason growth patterns using UNet, followed by cancer grading, and achieved performance comparable to pathologists [68]. Table 3 contains additional published examples utilizing various deep learning architectures and diverse datasets. A complete and extensive understanding of deep learning concepts and existing architectures can be found [17,69], while a specific application of deep learning in medical image analysis can be read [70,71,72]. To avoid bias in algorithm development, datasets should be truly representative, encompassing the range of data that would be expected in the real world [19], including both the expected range of tissue features (normal and pathological) and the expected variation in tissue and slide preparation between laboratories.

CNNs are difficult to train end-to-end because gigapixel WSIs are too large to fit in GPU memory, unlike many natural pictures evaluated in computer vision applications. A single WSI requires over terabytes of memory, yet high-end GPUs only give tens of gigabytes. Researchers have suggested alternatives such as partitioning the WSI into small sections (Figure 3) using only a subset or the full WSI compressed with semantic information preserved. Breaking WSI into little patches and placing them all in the GPU to learn everything takes too long; thus, picking patches to represent WSI is critical. For these reasons, randodmizing paches [86], selecting patches from region of interests [42], and randomly selecting patches among image clustering [87] were proposed. The multi-instance learning (MIL) method is then mostly employed in the patch aggregation step, which involves collecting several patches from a single WSI and learning information about the WSI as a result. Traditional MILs treat a single WSI as a basket, assuming that all patches contained within it have the same WSI properties. All patches from a cancer WSI, for example, are considered cancer patches. This method appears to be very simple, yet it is quite beneficial for cancer detection, and representation can be ensured if the learning dataset is large enough [88], which also provides a reason why various large datasets should be produced. If the learning size is insufficient, predicted patch scores are averaged, or classes that account for the majority of patch class predictions are estimated and used to represent the WSI. A more typical way is to learn patch weights using a self-attention mechanism, which uses patch encoding to calculate weighed sum of patch embeddings [89], with a higher weight for the patch that is closest to the ideal patch for performing a certain task for each model. Techniques such as max or mean pooling and certainty pooling, which are commonly utilized in CNNs, are sometimes applied here. There is an advantage to giving interpretability to pathologists using the algorithm because approaches such as self-attention can be presented in the form of a heatmap on a WSI based on patch weights.

### 3.3. Overview of Deep Learning Workflows

WSIs are flooding out of clinical pathology facilities around the world as a result of the development of CPATH, including publicly available datasets, which can be considered a desirable cornerstone for the development of deep learning because it means more data are available for research. However, as shown in some of the previous studies, the accuracy of performance, such as classification and segmentation by algorithms commonly expressed in the area under the curve (AUC), must be compared to pathological images manually annotated by humans in order to calculate the accuracy of the performance. In this way, supervised learning is a machine learning model that uses labeled learning data for algorithm learning and learns functions based on it, and it is the machine learning model most utilized in CPATH so far. According to the amount and type of data, object and purpose (whether the target is cancer tissue or substrate tissue and calculating the number of lymphocytes), it can be divided into qualitative and distinct or quantitative and continuous representations, expressed as ‘classification’ [90] and ‘regression’ [91], respectively. Because the model is constructed by simulating learning data, labeled data are crucial, and the machine learning model’s performance may vary. Unsupervised learning uses unlabeled images, unlike previous scenarios. This technology is closer to an AI since it helps humans collect information and build knowledge about the world. Except for the most basic learning, such as language character acquisition, we can identify commonalities by looking at applied situations and extending them to other objects. To teach young children to recognize dogs and cats, it is not required to exhibit all breeds. ‘Unsupervised learning’ can find and assess patterns in unlabeled data, divide them into groups, or perform data visualization in which specific qualities are compacted to two or three if there are multiple data characteristics or variables that are hard to see. A study built a complex tissue classifier for CNS tumours based on histopathologic patterns without manual annotation. It provided a framework comparable to the WHO [92], which was based on microscopic traits, molecular characteristics, and well-understood biology [93]. This study demonstrated that the computer can optimize and use some of the same histopathologic features used by pathologists to assist grouping on its own.

In CPATH, it is very important to figure out how accurate a newly made algorithm is, so there is still a lot of supervised learning. Unsupervised learning still makes it hard to keep up with user-defined tasks, but it has the benefit of being a very flexible way to build data patterns that are not predictable. It also lets us deal with changes we did not expect and allows us to learn more outside of the limits of traditional learning. It helps us understand histopathology images and acts as a guide for precision medicine [94].

Nonetheless, unsupervised learning is still underdeveloped in CPATH, and even after unsupervised learning, it is sometimes compared with labeled data to verify performance, making the purpose a little ambiguous. Bulten et al. classified prostate cancer and non-cancer pathology using clustering, but still had to verify the algorithm’s ability using manually annotated images, for example [95].

Currently, efforts are made to make different learning datasets by combining the best parts of supervised and unsupervised learning. This is done by manually labeling large groups of pathological images. Instead of manually labeling images, such as in the 2016 TUPAC Challenge, which was an attempt to build standard references for mitosis detection [96], “weakly supervised learning” means figuring out only a small part of an image and then using machine learning to fill in the rest. Several studies have shown that combining sophisticated learning strategies with weakly supervised learning methods can produce results that are similar to those of a fully supervised model. Since then, many more studies have been done on the role of detection and segmentation in histopathology images. “NuClick”, a CNN-based algorithm that won the LYON19 Challenge in 2019, showed that structures such as nuclei, cells, and glands in pathological images can be labeled quickly, consistently, and reliably [97], whereas ‘CAMEL’, developed in another study, only uses sparse image-level labels to produce pixel-level labels for creating datasets to train segmentation models for fully supervised learning [98].

## 4. Current Limitations and Challenges

Despite considerable technical advancements in CPATH in recent years, the deployment of deep learning algorithms in real clinical settings is still far from adequate. This is because, in order to be implemented into existing or future workflows, the CPATH algorithm must be scientifically validated, have considerable clinical benefit, and not cause harm or confuse people at the same time [99]. In this section, we will review the roadblocks to full clinical adoption of the CPATH algorithm, as well as what efforts are currently being made.

### 4.1. Acquiring Quality Data

It is critical that CPATH algorithms be trained with high-quality data so that they can deal with the diverse datasets encountered in real-world clinical practice. Even in deep learning, the ground truth should be manually incorporated into the dataset in order to train appropriate diagnostic contexts in supervised learning to classify, segment, and predict images based on it [100]. The ground truth can be derived from pathology reports grading patient outcomes or tumors, as well as scores assessed by molecular experiments, depending on the study’s goals, which are still determined by human experts and need a significant amount of manual labor to obtain a ‘correct’ dataset [12]. Despite the fact that datasets created by professional pathologists are of excellent quality, vast quantities are difficult to obtain due to the time, cost, and repetitive and arduous tasks required. As a result, publicly available datasets have been continuously created, such as the ones from TCGA or grand challenges, with the help of weakly supervised learning. Alternative efforts have recently been made to gather massive scales of annotated images by crowdsourcing online. Hughes et al. used a crowdsourced image presentation platform to demonstrate deep learning performance comparable to that of a single professional pathologist [101], while López-Pérez et al. used a crowdsourced deep learning algorithm to help a group of doctors or medical students who were not pathologists make annotations comparable to an expert in breast cancer images [102]. Crowdsourcing may generate some noise, but it shows that non-professionals of various skill levels could assist with pathological annotation and dataset generation. Obtaining quality data entails more than just obtaining a sufficient raw pathological image slide of a single disease from a patient or hospital; it also includes preparing materials to analyze and process the image in order to extract useful data for deep learning model training. By using strategies such as selecting patches with cells while excluding patches without cells from raw pictures, as demonstrated in Figure 4, collecting quality data may be made easier.

### 4.2. Data Variation

Platform diversity, integration, and interoperability represent yet another significant hurdle for the creation and use of AI tools [103]. Recent findings show that current AI models, when trained on insufficient datasets, even when utilizing precise and pixel-by-pixel labelling, can exhibit a 20% decline in performance when evaluated on independent datasets [88]. Deep learning-based algorithms have produced outstanding outcomes in image analysis applications, including digitized slide analysis. Deep learning-based systems face several technological problems, including huge WSI data, picture heterogeneity, and feature complexity. To achieve successful generalization properties, the training data must include a diverse and representative sample of the disease’s biological and morphological variability, as well as the technical variables introduced in the pre-analytical and analytical processes in the pathology department, as well as the image acquisition process [104]. A generic deep learning-based system for histopathology tissue analysis. The previously introduced framework is a series of strategies in the preprocessing-training-inference pipeline that showed improved efficiency and generalizability. Such strategies include an ensemble segmentation model, dividing the WSI into smaller overlapping patches, efficient inference algorithms, and a patch-based uncertainty estimation methodology [105,106]. Technical variability challenges can also be addressed by standardizing and preparing CPATH data to limit the effects of technical variability or to make the models robust to technical variability. Training the deep learning model on large and diverse datasets may lower the generalization error to some extent [107].

The amount and quality of input data determine the performance of the deep learning algorithm [108,109]. Although the size of datasets has been growing over the years with the development in CPATH, even if algorithms trained using learning datasets perform well on test sets, it is difficult to be certain that algorithms perform well on actual clinical encounters because clinical data come from significantly more diverse sources than studies. Similarly, when evaluating the performance of deep learning algorithms with a specific validation set for each grand challenge, it is also difficult to predict whether they will perform well in actual clinical practice. Color variation is a representative example of the variation of data. Color variation is caused by differences in raw materials, staining techniques used across different pathology labs, patient intervariability, and different slide scanners, which affect not just color but also overall data variation [110]. As a result, color standardization as an image preparation method has long been devised to overcome this problem in WSI. Because predefined template images were used for color normalization in the past, it was difficult to style transformation between different image datasets, but recent advances in generative adversarial networks (GAN) among deep learning artificial neural networks have allowed patches to be standardized without organizational changes. For example, using the cycle-GAN technique, Swiderska-Chadaj et al. reported an AUC of 0.98 and 0.97 for two different datasets constructed from prostate cancer WSIs [72,111]. While efforts are being made to reduce variation and create well-defined standardized data, such as color standardization and attempts to establish global standards for pathological tissue processing, staining, scanning, and digital image processing, data augmentation techniques are also being used to create learning datasets with as many variations as possible in order to learn the many variations encountered in real life. Not only the performance of the CPATH algorithm but also many considerations such as cost and explainability should be thoroughly addressed when deciding which is more effective for actual clinical introduction.

### 4.3. Algorithm Validation

Several steps of validation are conducted during the lengthy process of developing a CPATH algorithm in order to test its performance and safety. To train models and evaluate performance, CPATH studies on typical supervised algorithms separate annotated data into individual learning datasets and test datasets, the majority of which employ datasets with features fairly similar to those of learning datasets in the so-called ‘internal verification’ stage. Afterwards, through so-called ‘external validation’, which uses data for tests that have not been used for training, it is feasible to roughly evaluate if the algorithm performs well with the data it would encounter in real clinical practice [15]. However, simply because the CPATH algorithm performed well at this phase, it is hard to ascertain whether it will function equally well in practical practice [112]. While many studies on the CPATH algorithm are being conducted, most studies use autonomous standards due to a lack of established clinical verification standards and institutional validation. Even if deep learning algorithms perform well and are employed with provisional permission, it is difficult to confirm that their performance exhibits the same confirmed effect when the algorithm is upgraded in the subsequent operation process. Efforts are being made to comprehend and compare diverse algorithms regardless of research techniques, such as the construction of a complete and transparent information reporting system called TRIPOD-AI in the prediction model [113].

Finally, it should be noted that the developed algorithm does not result in a single performance but rather continues within the patient’s disease progress and play an auxiliary role in decision-making; thus, relying solely on performance as a ratification metric is not ideal. This suggests that, in cases where quality measure for CPATH algorithm performance is generally deemed superior to or comparable to pathologists, it should be defined by examining the role of algorithms in the whole scope of disease progression in a patient in practice [114]. This is also linked to the solution of the gold-standard paradox [14]. This is a paradox which may ariase during the segmentation model’s quality control, where pathologists are thought to be the most competent in pathological picture analysis, but algorithmic data are likely to be superior in accuracy and reproducibility. This paradox may alternatively be overcome by implementing the algorithm as part of a larger system that tracks the patient’s progress and outcomes [12].

### 4.4. Regulatory Considerations

One of the most crucial aspects for deep learning algorithms to be approved by regulatory agencies in order to use AI in clinical practice is to understand how it works, as AI is sometimes referred to be a “black box” because it is difficult for humans to comprehend exactly what it does [114]. Given the difficulty of opening up deep learning artificial neural networks and their limited explainability due to the difficulty of understanding how countless parameters interact at the same time, more reliable and explainable models for complex and responsible behaviors for diagnosis and treatment decisions and prediction are required [115]. As a result, attempts have been made to turn deep learning algorithms into “glass boxes” by clarifying the input and calculating the output in a way that humans can understand and analyze [116,117,118].

The existing regulatory paradigm is less adequate for AI since it requires rather small infrastructure and little human interaction, and the level of progress or results are opaque to outsiders, so potential dangers are usually difficult to identify [119]. Thus far, the White House has issued a memorandum on high-level regulatory principles for AI in all fields in November 2020 [120], the European Commission issued a similar white paper in February 2020 [121], and UNESCO made a global guideline on AI ethics in November 2021 [122], but these documents unfortunately do not provide a very detailed method to operate artificial intelligence in the context of operations. Because artificial intelligence is generally developed in confined computer systems, progress has been made outside of regulatory environments thus far, and regulatory uncertainty can accelerate development while also fueling systemic dangers at the same time. Successful AI regulations, as with many new technologies, are expected to be continuously problematic in the future, as regulations and legal rules will still lag behind developing technological breakthroughs [123]. Self-regulation in industrial settings can be theoretically beneficial and is already in use [124], but it has limitations in practice because it is not enforced. Ultimately, a significant degree of regulatory innovation is required to develop a stable AI environment. The most crucial issue to consider in this regard is that, in domains such as health care, where even a slight change can have a serious influence, regulations of AI should be built with the consideration of the overall impact on humans rather than making arbitrary decisions alone.

## 5. Novel Trends in CPATH

### 5.1. Explainable AI

Because most AI algorithms have unclear properties due to their complexity and often lacking robustness, there are substantial issues with AI trust [125]. Furthermore, there is no agreement on how pathologists should include computational pathology systems into their workflow [126]. Building computational pathology systems with explainable artificial intelligence (xAI) methods is a strong substitute for opaque AI models to address these issues [127]. Four categories of needs exist for the usage of xAI techniques and their application possibilities [128]: (1) Model justification: to explain why a decision was made, particularly when a significant or unexpected decision is created, all with the goal of developing trust in the model’s operation; (2) Model controlling and debugging: to avoid dangerous outcomes. A better understanding of the system raises the visibility of unknown defects and aids in the rapid identification and correction of problems; (3) Model improving: When a user understands why and how a system achieved a specific result, he can readily modify and improve it, making it wiser and possibly faster. Understanding the judgments created by the AI model, in addition to strengthening the explanation-generating model, can improve the overall work process; (4) Knowledge discovery: One can discover new rules by seeing the appearance of some invisible model results and understanding why and how they appeared. Furthermore, because AI entities are frequently smarter than humans, it is possible to learn new abilities by understanding their behavior.

Recent studies in breast pathology xAI quickly presented the important diagnostic areas in an interactive and understandable manner by automatically previewing tissue WSIs and identifying the regions of interest, which can serve pathologists as an interactive computational guide for computer-assisted primary diagnosis [127,129]. An ongoing study is being done to determine which explanations are best for artificial intelligence development, application, and quality control [130], which explanations are appropriate for situations with high stakes [115], and which explanations are true to the explained model [131].

With the increasing popularity of graph neural networks (GNNs), their application in a variety of disciplines requires explanations for scientific or ethical reasons in medicine [132]. This makes it difficult to define generalized explanation methods, which are further complicated by heterogeneous data domains and graphs. Most explanations are therefore model- and domain-specific. GNN models can be used for node labeling, link prediction, and graph classification [133]. While most models can be used for any of the above tasks, defining and generating explanations can affect how a GNN xAI model is structured. However, the power of these GNN models is limited by their complexity and the underlying data complexity, although most, if not all, of the models can be grouped under the augmented paradigm [134]. Popular deep learning algorithms and explainability techniques based on pixel-wise processing ignore biological elements, limiting pathologists’ comprehension. Using biological entity-based graph processing and graph explainers, pathologists can now access explanations.

### 5.2. Ethics and Security

AI tool creation must take into account the requirement for research and ethics approval, which is typically necessary during the research and clinical trial stages. Developers must follow the ethics of using patient data for research and commercial advantages. Recognizing the usefulness of patient data for research and the difficulties in obtaining agreement for its use, the corresponding institution should establish a proper scheme to provide individual patients some influence over how their data are used [103]. Individual institutional review boards may have additional local protocols for permitting one to opt out of data use for research, and it is critical that all of these elements are understood and followed throughout the design stage of AI tool creation [104]. There are many parallels to be found with the AI development pipeline; while successful items will most likely transit through the full pathway, supported by various resources, many products will, however, fail at some point. Each stage of the pipeline, including the justification of the tool for review and being recommended for usage in clinical guidelines, can benefit from measurable outcomes of success in order to make informed judgments about which products should be promoted [135]. This usually calls for proof of cost or resource savings, quality improvements, and patient impact and is thus frequently challenging to demonstrate, especially when the solution entails major transformation and process redesign.

Whether one uses a cloud-based AI solution for pathology diagnostics depends on a number of things, such as the preferred workflow, frequency of instrument use, software and hardware costs, and whether or not the IT security risk group is willing to allow the use of cloud-based solutions. Cloud-based systems must include a business associate’s agreement, end-to-end encryption, and unambiguous data-use agreements to prevent data breaches and inappropriate use of patient data [21].

## 6. Conclusions and Future Directions

AI currently has enormous potential to improve pathology practice by reducing errors, improving reproducibility, and facilitating expert communication, all of which were previously difficult with microscopic glass slides. Recent trends of AI applicaion should be affordable, practical, interoperable, explainable, generalizable, manageable, and reimbursable [21]. Many researchers are convinced that AI in general and deep learning in particular could help with many repetitive tasks using digital pathology because of recent successes in image recognition. However, there are currently only a few AI-driven software tools in this field. As a result, we believe pathologists should be involved from the start, even when developing algorithms, to ensure that these eagerly anticipated software packages are improved or even replaced by AI algorithms. Despite popular belief, AI will be difficult to implement in pathology. AI tools are likely to be approved by regulators such as the Food and Drug Administration.

The quantitative nature of CPATH has the potential to transform pathology laboratory and clinical practices. Case stratification, expedited review and annotation, and the output of meaningful models to guide treatment decisions and predict patterns in medical fields are all possibilities. The pathology community needs more research to develop safe and reliable AI. As clinical AI’s requirements become clearer, this gap will close. AI in pathology is young and will continue to mature as researchers, doctors, industry, regulatory agencies, and patient advocacy groups innovate and bring new technology to health care practitioners. To accomplish its successful application, robust and standardized computational, clinical, and laboratory practices must be established concurrently and validated across multiple partnering sites.

## Figures and Tables

**Figure 1 diagnostics-12-02794-f001:**
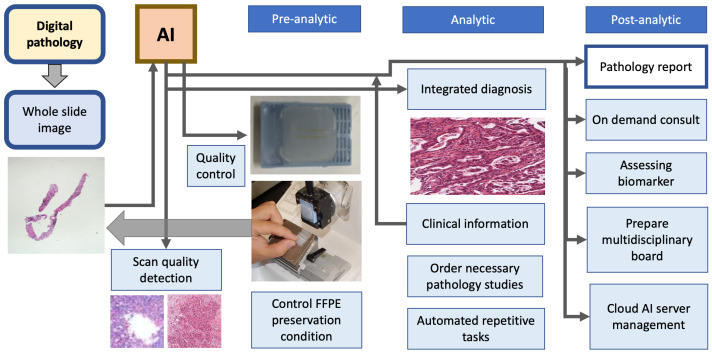
Embedding AI into pathology department workflow. The digital pathology supplies whole-slide images to artificial intelligence, which performs quality control of pre-analytic phase, analytic phase and post-analytic phase of pathology laboratory process.

**Figure 2 diagnostics-12-02794-f002:**
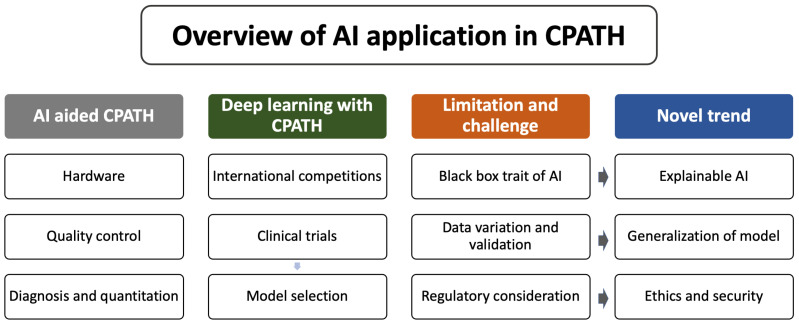
Requirement for clinical applications of artificial intelligence with CPATH.

**Figure 3 diagnostics-12-02794-f003:**
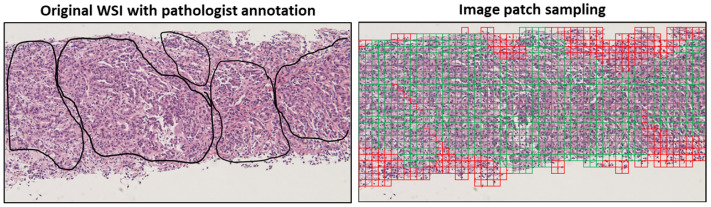
Images are divided into small patches obtained from tissue of WSI, which are subsequently prepared to have semantic features extracted from each patch. Green tiles indicate tumor region; red tiles indicate non-tumor region. Images from Yeouido St. Mary’s hospital.

**Figure 4 diagnostics-12-02794-f004:**
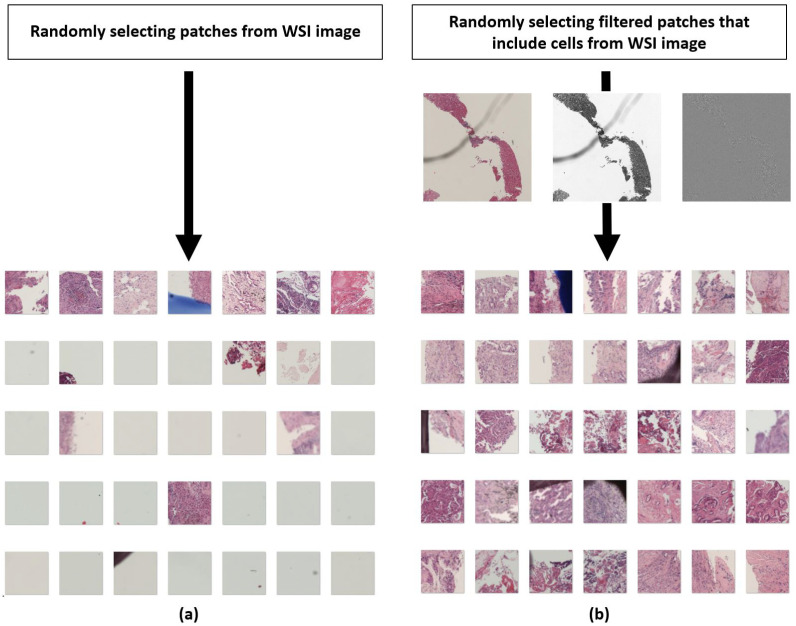
(**a**) Random sampling of 100 patches selected arbitrarily from an WSI image. (**b**) Random sampling of 100 patches after application of Laplace filter (which highlights areas with great changes) from WSI image. Images from Yeouido St. Mary’s Hospital.

**Table 1 diagnostics-12-02794-t001:** Computational pathology definitions.

Terms	Definition
Artificial intelligence (AI)	The broadest definition of computer science dealing with the ability of a computer to simulate human intelligence and perform complicated tasks.
Computational pathology (CPATH)	A branch of pathology that involves computational analysis of a broad array of methods to analyze patient specimens for the study of disease. In this paper, we focus on the extraction of information from digitized pathology images in combination with their associated metadata, typically using AI methods such as deep learning.
Convolutional neural networks (CNN)	A form of deep neural networks with one or more convolutional layers and various different layers that can be trained using the backpropagation algorithm and which is suitable for learning 2D data such as images.
Deep learning	A subclassification of machine learning that imitates a logical structure similar to how people conclude using a layered algorithm structure called an artificial neural network.
Digital pathology	An environment in which traditional pathology analysis utilizing slides made of cells or tissues is converted to a digital environment using a high-resolution scanner.
End-to-end training	An opposite concept of feature-crafted methods in a machine learning model, a method which learns the ideal value simultaneously rather than sequentially using only one pipeline. It works smoothly when the dataset is large enough.
Ground truth	A concept of a dataset’s ‘true’ category, quantity, or label that serves as direction to an algorithm in the training step. The ground truth varies from the patient- or slide-level to objects or areas within the picture, depending on the objective.
Image segmentation	A technique for classifying each region into a semantic category by decomposing an image to the pixel level.
Machine learning	An artificial intelligence that parses data, learns from it, and makes intelligent judgments based on what it has learned.
Metadata	A type of data that explains other data. A single histopathology slide image in CPATH may include patient disease, demographic information, previous treatment records and medical history, slide dyeing information, and scanner information as metadata.
Whole-slide image (WSI)	An whole histopathological glass slide digitized at microscopic resolution as a digital representation. Slide scanners are commonly used to create these complete slide scans. A slide scan viewing platform allows for image examination similar to that of a regular microscope.

**Table 2 diagnostics-12-02794-t002:** Examples of grand challenges held in CPATH.

Challenge	Year	Staining	Challenge Goal	Dataset
GlaS challenge [55]	2015	H&E	Segmentation of colon glands of stage T3 and T4 colorectal adenocarcinoma	Private set—165 images from 16 WSIs
CAMELYON16 [56]	2016	H&E	Evaluation of new and current algorithms for automatic identification of metastases in WSIs from H&E-stained lymph node sections	Private set—221 images
TUPAC challenge [57]	2016	H&E	Prediction of tumor proliferation scores and gene expression of breast cancer using histopathology WSIs	821 TCGA WSIs
BreastPathQ [58]	2018	H&E	Development of quantitative biomarkers to determinate cancer cellularity of breast cancer from H&E-stained WSIs	Private set—96 WSIs
BACH challenge [59]	2018	H&E	Classification of H&E-stained breast histopathology images and performing pixel-wise labeling of WSIs	Private set—40 WSIs and 500 images
LYON19 [60]	2019	IHC	Provision of a dataset as well as an evolution platform for current lymphocyte detection algorithms in IHC-stained images	LYON19 test set containing 441 ROIs
DigestPath [61]	2019	H&E	Evaluation of algorithms for detecting signet ring cells and screening colonoscopy tissue from histopathology images of the digestive system	Private set—127 WSIs
HEROHE ECDP [62]	2020	H&E	Evaluation of algorithms to discriminate HER2-positive breast cancer specimens from HER2-negative breast cancer specimens with high sensitivity and specificity only using H&E-stained slides	Private set—359 WSIs
MIDOG challenge [63]	2021	H&E	Detection of mitotic figures from breast cancer histopathology images scanned by different scanners to overcome the ‘domain-shift’ problem and improve generalization	Private set—200 cases
CoNIC challenge [64]	2022	H&E	Evaluation of algorithms for nuclear segmentation and classification into six types, along with cellular composition prediction	4981 patches
ACROBAT [65]	2022	H&E, IHC	Development of WSI registration algorithms that can align WSIs of IHC-stained breast cancer tissue sections with corresponding H&E-stained tissue regions	Private dataset—750 cases consisting of 1 H&E and 1–4 matched IHC

**Table 3 diagnostics-12-02794-t003:** Summary of recent convolutional neural network models in pathology image analysis.

Publication	Deep Learning	Input	Training Goal	Dataset
Zhang et al. [73]	CNN	WSI	Diagnosis of bladder cancer	TCGA and private—913 WSIs
Shim et al. [74]	CNN	WSI	Prognosis of lung cancer	Private—393 WSIs
Im et al. [75]	CNN	WSI	Diagnosis of brain tumor subtype	private—468 WSIs
Mi et al. [76]	CNN	WSI	Diagnosis of breast cancer	private dataset—540 WSIs
Hu et al. [77]	CNN	WSI	Diagnosis of gastric cancer	private—921 WSIs
Pei et al. [78]	CNN	WSI	Diagnosis of brain tumor classification	TCGA—549 WSIs
Salvi et al. [79]	CNN	WSI	Segmentation of normal prostate gland	Private—150 WSIs
Lu et al. [80]	CNN	WSI	Genomic correlation of breast cancer	TCGA and private—1157 WSIs
Cheng et al. [81]	CNN	WSI	Screening of cervical cancer	Private—3545 WSIs
Kers et al. [82]	CNN	WSI	Classification of transplant kidney	Private—5844 WSIs
Zhou et al. [83]	CNN	WSI	Classification of colon cancer	TCGA—1346 WSIs
Hohn et al. [84]	CNN	WSI	Classification of skin cancer	Private—431 WSIs
Wang et al. [45]	CNN	WSI	Prognosis of gastric cancer	Private—700 WSIs
Shin et al. [85]	CNN, GAN	WSI	Diagnosis of ovarian cancer	TCGA—142 WSIs

Abbreviation: CNN, convolutional neural network; WSI, whole-slide image; TCGA, The Cancer Genome Atlas.

## Data Availability

Data sharing is not applicable to this article.

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
