# Peer review of "Application of Artificial Intelligence in Pathology: Trends and Challenges"

_diagnostics, 2022, doi:10.3390/diagnostics12112794_

Round 1
Reviewer 1 Report
Abstract is week, looks like an introduction, moreover try to cover the aspects which are covered in the paper in different sections and why?
Introduction: lines 15 to 20 are about AI and perceptron, really required, I think no?
Rather start introduction for your paper title to defend.
Pages 3 and 4 are written well.
Table offers better information for literature, table 2 and table 3.
Work is a bit difficult to understand as an organization, so suggestion is to keep an outline of whole work as an graphical abstract for better understanding of different sections.
Conclusion and Future work is not sufficient and gives an impression as incomplete.
More recent references can be added in the literature around AI and ML application and its areas.
HRDEL: High Ranking Deep Ensemble Learning-based Lung Cancer Diagnosis Model
FETCH: A Deep Learning-Based Fog Computing and IoT Integrated Environment for Healthcare Monitoring and Diagnosis
Author Response
Abstract is week, looks like an introduction, moreover try to cover the aspects which are covered in the paper in different sections and why?
Thank you for the precious comment, we revised our abstract and reduced redundancy.
Introduction: lines 15 to 20 are about AI and perceptron, really required, I think no?
Thank you for the valuable comment. We omitted the contents
Rather start introduction for your paper title to defend.
Thank you for the precious comment, we changed the starts of introduction.
Pages 3 and 4 are written well.
Table offers better information for literature, table 2 and table 3.
Work is a bit difficult to understand as an organization, so suggestion is to keep an outline of whole work as an graphical abstract for better understanding of different sections.
Thank you for the precious comment.
- we re-organized the whole manuscript and changed subjects accordingly.
- We added figure 2 for better understanding of manuscript content
Conclusion and Future work is not sufficient and gives an impression as incomplete.
- Thank you for the important comment. We revised conclusion.
More recent references can be added in the literature around AI and ML application and its areas.
Thank you for the valuable comment. We added below references
HRDEL: High Ranking Deep Ensemble Learning-based Lung Cancer Diagnosis Model
FETCH: A Deep Learning-Based Fog Computing and IoT Integrated Environment for Healthcare Monitoring and Diagnosis
Reviewer 2 Report
This review article entitled “Application of Artificial Intelligence in Pathology: Trends and Challenges” can potentially provide some useful information about the recent progresses and new tools in the field of AI-assisted pathology diagnosis and to help improve the diagnostic accuracy of traditional pathology. Having said that, this review needs some major work to improve its quality. The following are my suggestions.
1. It is difficult to read this manuscript. There are many very long sentences. For example, at the beginning of Introduction, it says “The perceptron, which introduced the main concepts of neural networks and neural networks underlying artificial intelligence, was introduced in the mid-20th century, but it wasn’t until AlexNet won the ImageNet large-scale image recognition competition in 2012 that deep learning became a full-fledged global research topic that is now being actively studied in all fields[1]…Although computer-aided diagnostics has long been studied in the field of medical image interpretation, it has not been widely used in clinical practice because computer-aided diagnostics systems based on traditional machine learning systems were found to produce more false positives than physicians, thus failing to outperform or improve their work efficiency in real practice [7–9], which can be seen as a result of the fundamental structural limitations of artificial intelligence at the time, when researchers struggled to solve the complex problem with multi-layer perceptron due to the so-called ’vanishing gradient’ problem, which is a phenomenon in which data vanishes during back propagation learning.” There are many long sentences like these. Please break those long sentences into simple individual sentences, so they can be easily understood.
2. The manuscript is difficult to follow for non-specialists/pathologist. It is better to make a rearrangement of the figures (revising the figs 1 and 2) to introduce the concepts, terms/elements and challenges first, and then describe recent developments and solutions to those challenges.
3. In the figure 2, what does the “competition and challenge” mean there? The supervised, unsupervised or weakly supervised machine learning should be moved to the challenges of data analysis section.
4. Since high resolution/efficiency scanners and their costs are critical for implementation of AI-assisted digital pathology, and the recent trends toward spatial molecular imaging, the authors may want to include some recent developments in these fields.
Author Response
This review article entitled “Application of Artificial Intelligence in Pathology: Trends and Challenges” can potentially provide some useful information about the recent progresses and new tools in the field of AI-assisted pathology diagnosis and to help improve the diagnostic accuracy of traditional pathology. Having said that, this review needs some major work to improve its quality. The following are my suggestions.
- It is difficult to read this manuscript. There are many very long sentences. For example, at the beginning of Introduction, it says “The perceptron, which introduced the main concepts of neural networks and neural networks underlying artificial intelligence, was introduced in the mid-20th century, but it wasn’t until AlexNet won the ImageNet large-scale image recognition competition in 2012 that deep learning became a full-fledged global research topic that is now being actively studied in all fields[1]…Although computer-aided diagnostics has long been studied in the field of medical image interpretation, it has not been widely used in clinical practice because computer-aided diagnostics systems based on traditional machine learning systems were found to produce more false positives than physicians, thus failing to outperform or improve their work efficiency in real practice [7–9], which can be seen as a result of the fundamental structural limitations of artificial intelligence at the time, when researchers struggled to solve the complex problem with multi-layer perceptron due to the so-called ’vanishing gradient’ problem, which is a phenomenon in which data vanishes during back propagation learning.” There are many long sentences like these. Please break those long sentences into simple individual sentences, so they can be easily understood.
- Thank you for the precious comment, we re-organized the whole manuscript and changed subjects accordingly. We revised to eliminate redundancy in the contents
- The manuscript is difficult to follow for non-specialists/pathologist. It is better to make a rearrangement of the figures (revising the figs 1 and 2) to introduce the concepts, terms/elements and challenges first, and then describe recent developments and solutions to those challenges.
- Thank you for the comment, we added figure 1 to improve the understanding of readers
- In the figure 2, what does the “competition and challenge” mean there? The supervised, unsupervised or weakly supervised machine learning should be moved to the challenges of data analysis section.
- thank you for the comment, we revised figure 2 and mentioned contents.
- Since high resolution/efficiency scanners and their costs are critical for implementation of AI-assisted digital pathology, and the recent trends toward spatial molecular imaging, the authors may want to include some recent developments in these fields.
- thank you for the comments, we added above mentioned contents in our manuscript.
Reviewer 3 Report
A well-prepared summary of the use of tools based on artificial intelligence so far.
Author Response
Thank you for the precious comments. we added figure 1 to improve the reader's understanding and revised abstract in addition to re-organize the whole manuscript and eliminate redundancy.
Round 2
Reviewer 1 Report
Comments are handled
Reviewer 2 Report
Most of my concerns/questions have been addressed, and this version is significantly better and much easier to read than the previous one.